# Multiple proteases are involved in mesothelin shedding by cancer cells

Xiufen Liu[1], Alexander Chan[1,3], Chin-Hsien Tai[1], Thorkell Andresson[2] & Ira Pastan [1✉]

Mesothelin (MSLN) is a lineage restricted cell surface protein expressed in about 30% of human cancers and high MSLN expression is associated with poor survival in several different cancers. The restricted expression of MSLN in normal tissue and its frequent expression in cancers make MSLN an excellent target for antibody-based therapies. Many clinical trials with agents targeting MSLN have been carried out but to date none of these agents have produced enough responses to obtain FDA approval. MSLN shedding is an important factor that may contribute to the failure of these therapies, because shed MSLN acts as a decoy receptor and allows release of antibodies bound to cell-surface MSLN. We have investigated the mechanism of shedding and show here that members of the ADAM, MMP and BACE families of proteases all participate in shedding, that more than one protease can produce shedding in the same cell, and that inhibition of shedding greatly enhances killing of cells by an immunotoxin targeting MSLN. Our data indicates that controlling MSLN shedding could greatly increase the activity of therapies that target MSLN.

[1] Laboratory of Molecular Biology, Center for Cancer Research, National Cancer Institute, National Institutes of Health, 37 Convent Drive, Bldg 37, Bethesda, MD 20892, USA. [2] Protein Characterization Laboratory, Cancer Research Technology Program, Frederick National Laboratory for Cancer Research, Leidos Biomedical Research, Inc., 8560 Progress Dr, Rm C1017, Frederick, MD 21701, USA. [3]Present address: Department of Bioengineering, School of Engineering and Applied Science, University of Pennsylvania, 210 South 33rd Street, Philadelphia, PA 19104, USA. ✉email: pastani@mail.nih.gov

Mesothelin (MSLN) is a GPI-anchored membrane protein that is synthesized as a 70 kDa precursor protein and is proteolytically cleaved, generating a 30 kDa polypeptide known as megakaryocyte potentiating factor that is released from the cell and a 40–45 kDa glycoprotein (MSLN)[1–6] that is connected to the cell surface by phosphatidyl inositol[7,8]. MSLN binds to MUC16 and has a role in cell adhesion[9,10] and tumor progression[11]. MSLN is shed from the cell surface generating a pool of free antigen in pleural fluid, ascites, and in blood. Serum MSLN is used as a screening biomarker for cancer patients and to monitor responses[12,13]. Unfortunately, shedding is an impediment to antibody-based therapies, because it provides a sink within tumors that reduces their efficiency and also promote their release from the cell surface before they can kill the cell[6].

Sheddases are proteases that cleave membrane-bound protein substrates close to or within the cell membrane and release the ectodomain from the cell. The best known sheddases are members of the ADAM family[14–16]. Substrates for ADAMs include growth factors, cytokines, chemokines, and adhesion molecules. ADAM17 also known as TACE was shown to mediate MSLN shedding in A431 cells[6]. However, TACE knockdown only reduced MSLN shedding by 50% indicating other sheddases have a role in shedding. To understand MSLN shedding in more depth, we determined the cut sites generated when MSLN is shed in different cancer cell lines. Using RNAseq data to determine sheddase expression, we knocked down RNAs encoding various sheddases to find which are important for shedding. Our results show that MSLN is cleaved at many sites close to the cell membrane, that ADAM10, ADAM17, BACE2, BACE1, and MMP15 all have a role in MSLN shedding, and that more than one sheddase can catalyze MSLN release in the same cell line.

## Results and discussion

To determine the sites at which MSLN is cut, we studied five different cancer cell lines: KLM1 (pancreatic), KB31 (cervical), OVCAR8 (ovarian), A431/H9 cells (epidermoid carcinoma transfected with MSLN), and RH16, a primary mesothelioma cell line. Using culture medium, we immunoprecipitated MSLN with an antibody to the amino terminus and identified the product by immuno-blotting and Coomassie Blue staining. Figure 1a shows an SDS gel in which MSLN appears as a 40–45 kDa broad band typical of a glycosylated protein. A431 cells do not express MSLN. Because a single band with a molecular weight close to that of full-length MSLN was detected, we conclude that the major cleavage site is near the cell membrane. Figure 1b lists the most C-terminal MSLN peptide identified by mass spectrometry (for full-length sequence of the peptide see Supplementary Table 1). It was previously reported that MSLN from human ascites, pleural fluid, or the A431/H9 cell line is cut at two locations near the C terminus[6,17]. One cleavage is at residue 586 between Y and L, and the other is at residue 591 between L and S (Fig. 1b, cut sites 3 and 6). We have now identified several other cut sites (Fig. 1b). All the cut sites are located close together and close to the plasma membrane. These results establish that the region close to the cell membrane is the principal location of cutting by sheddases. The multiplicity of sites raises the possibility that different enzymes are responsible for shedding in different cells.

## ADAM family

To identify the proteases responsible for shedding, we analyzed several RNAseq databases to determine which sheddases are expressed in our cell lines (https://depmap.org/portal/ccle). In KLM1, KB31, and OVCAR8 cells, several members of the ADAM family are expressed, specifically ADAM9, 10, 15, and 17 (Supplementary Table 2). ADAM17 was previously shown to be important for shedding of MSLN in A431/H9 cells[6]. Figure 2a, b (full image Supplementary Fig. 1) show that lowering ADAM17

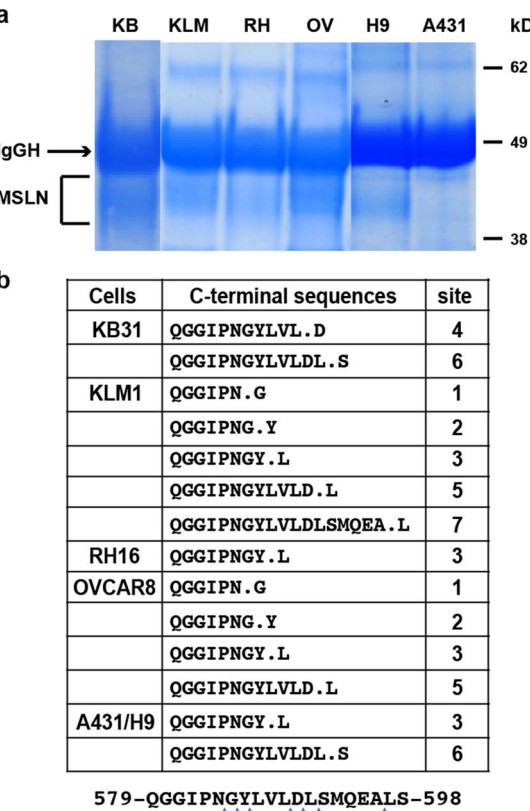

**Fig. 1 Identification of C-terminal sequence of MSLN. a** MSLN was immunoprecipitated from cell culture media using mouse monoclonal Ab MN. The areas labeled MSLN were cut out and analyzed by LC-MS. A431 cells do not express MSLN. KB for KB31, KLM for KLM1; RH for RH16; OV for OVCAR8; H9 for A431/H9, an MSLN stably transfected line. IgG H, heavy chain. **b** C-terminal sequences of MSLN from various cell lines. A431/H9 sequences are from published data[6]. Arrows indicate cut sites.

expression did not decrease MSLN secretion in KB31 but did decrease shedding in KLM1 cells by 45%. These data identify ADAM17 as a sheddase for KLM1 and indicate that other proteases must be involved in shedding. ADAM9, 10, and 15 also have protease activity[14] and are expressed in these cell lines (Supplementary Table 2). Figure 2c, d shows that ADAM10 knockdown diminished shedding in KLM1 and OVCAR8 cells while knockdown of both ADAM17 and ADAM10 produced a greater decrease in shedding than each alone Fig. 2e, f. However, knockdown of ADAM9 and ADAM15 did not lower MSLN shedding (Supplementary Figs. 2–4). Knockdown of ADAM9, 10, and 15 did not decrease KB31 cell shedding (Supplementary Fig. 2).

**MMPs**. MT-MMPs (membrane-type matrix metalloproteinases) are a subgroup of the MMP family that can also act as canonical sheddases[16]. MMP14, 15, 17, 24, and 25 are all MT-MMPs. We found that MMP14 is highly expressed in KLM1 cells and MMP15 is highly expressed in KB31, KLM1, and OVCAR8 cells (Supplementary Table 2). We observed that knockdown of MMP15 reduced MSLN expression in KB31 cells (Fig. 3a, b). Knockdown of MMP14 or MMP15 did not reduce MSLN shedding in KLM1 cells (Fig. 3c, d) or OVCAR8 cells (Supplementary Fig 5), but slightly increased shedding.

**BACE1 and BACE2**. BACE1 and BACE2 can cleave amyloid precursor protein (APP) and have been extensively studied in

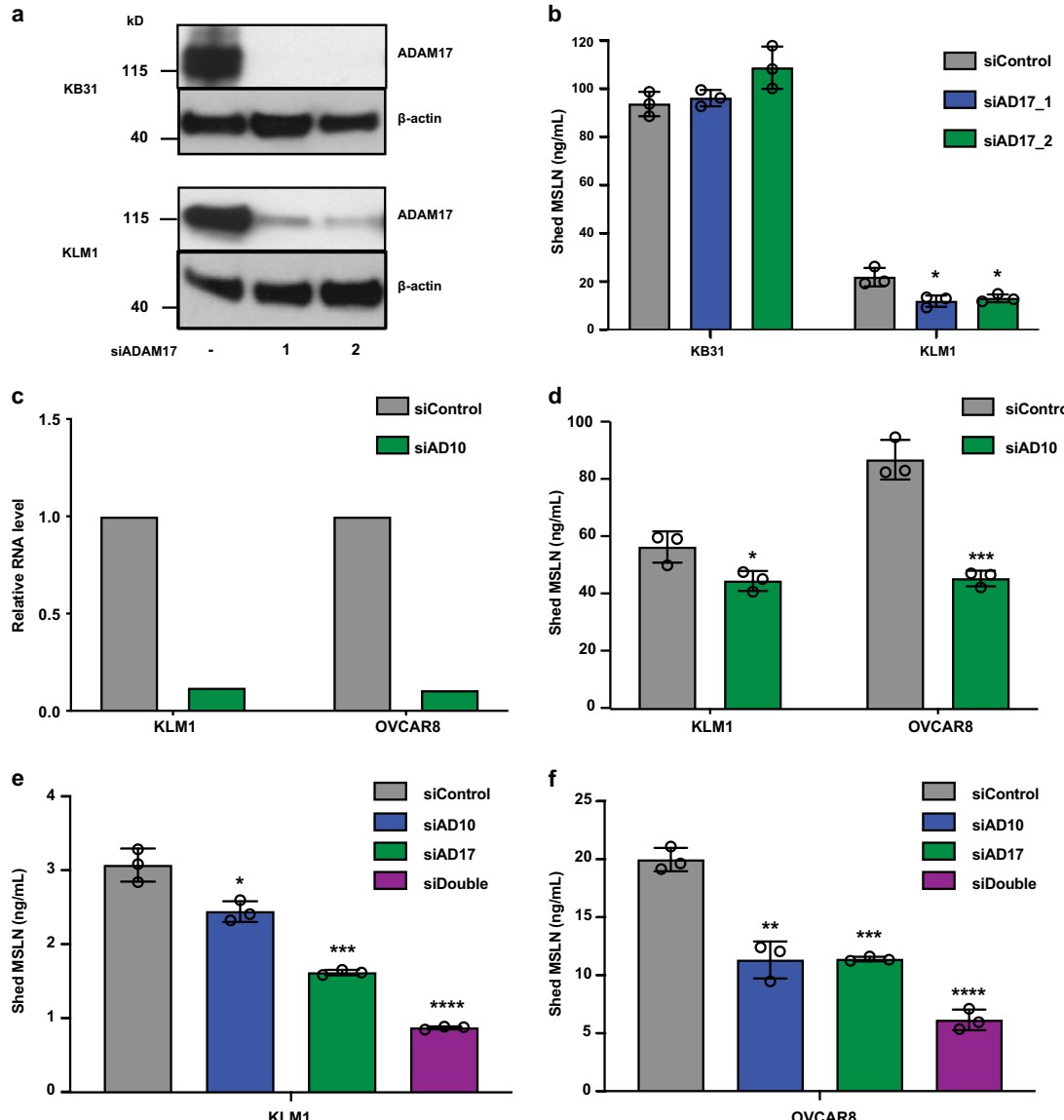

**Fig. 2 Knockdown of ADAM10 and ADAM17 decrease MSLN shedding in KLM1 and OVCAR8 cells, but not in KB31. a** Cells were transfected with siRNAs and lysates were analyzed by western blot using anti-ADAM17 Ab. Actin is a loading control (full blots shown in Supplementary Fig. 1). **b** Shed MSLN in cell culture media 48–72 h post-transfection from samples in **a** ($P = 0.018$ and $0.022$, $n = 3$). **c, d** ADAM10 siRNA or control were transfected into KLM1 and OVCAR8 cells. ADAM10 RNA was analyzed by real-time PCR after 48 h (**c**) and shed MSLN was measured 48–72 h post-transfection (**d** KLM1, $P = 0.034$; OVCAR8, $P = 0.0006$, $n = 3$). **e, f** Shed MSLN in KLM1 (**e** $P = 0.0146$, $0.00037$, and $0.00007$, $n = 3$) or OVCAR8 (**f** $P = 0.0014$, $0.00013$, $0.00006$, $n = 3$) culture media was analyzed 48–72 h post-transfection of control, ADAM10, ADAM17, and double knockdown siRNAs. The shed MSLN levels were normalized by WST-8 assay for cell growth. Statistical significance is noted by asterisks (*$P < 0.05$, **$P < 0.01$, ***$P < 0.001$, ****$P < 0.0001$).

Alzheimer's disease[18]. They have also been implicated in processing other proteins[16,19,20]. We find that BACE2 and BACE1 RNAs are expressed in many cell lines (Supplementary Table 2). To determine if BACE2 protein is also expressed, we did western blots and observed BACE2 expression in KB31, MS751, T3M4, and AsPC1 cells (Supplementary Fig. 6). To determine if BACE2 is involved in shedding, two siRNAs targeting human BACE2 were transfected into KB31 cells. Western blots show that when BACE2 protein is knocked down (Fig. 4a, full image Supplemental Fig. 7), total cell-associated MSLN is greatly increased (Fig. 4b), cell surface MSLN is increased as measured by flow cytometry (Fig. 4c), and shed MSLN is substantially decreased (Fig. 4d). These data indicate that in KB31 cells, BACE2 knockdown increases cell-associated MSLN by reducing MSLN shedding.

Since BACE1 RNA is also expressed in KB31 cells, albeit at a lower level than BACE2 (Supplementary Table 2), we knocked them down (Fig. 4e) and found that knockdown of BACE1 or BACE2 lowered shed MSLN levels to the same extent (Fig. 4f). However, knocking down both did not further lower MSLN shedding.

Because BACE2 is expressed more abundantly than BACE1 in our cell line panel (Supplementary Table 2), we examined the role of BACE2 in a cervical cancer line, MS751, and two pancreatic cancer lines, AsPC1 and T3M4. Figure 5a shows that siRNAs for BACE2 lower BACE2 protein levels (full image Supplemental Fig. 8). Figure 5b shows that in the medium of these three lines MSLN is significantly decreased. In MS751 there is a 60% decrease, in T3M4 a 55% decrease, and in AsPC1 a 23% decrease. We also knocked down BACE1 in KLM1 and BACE1

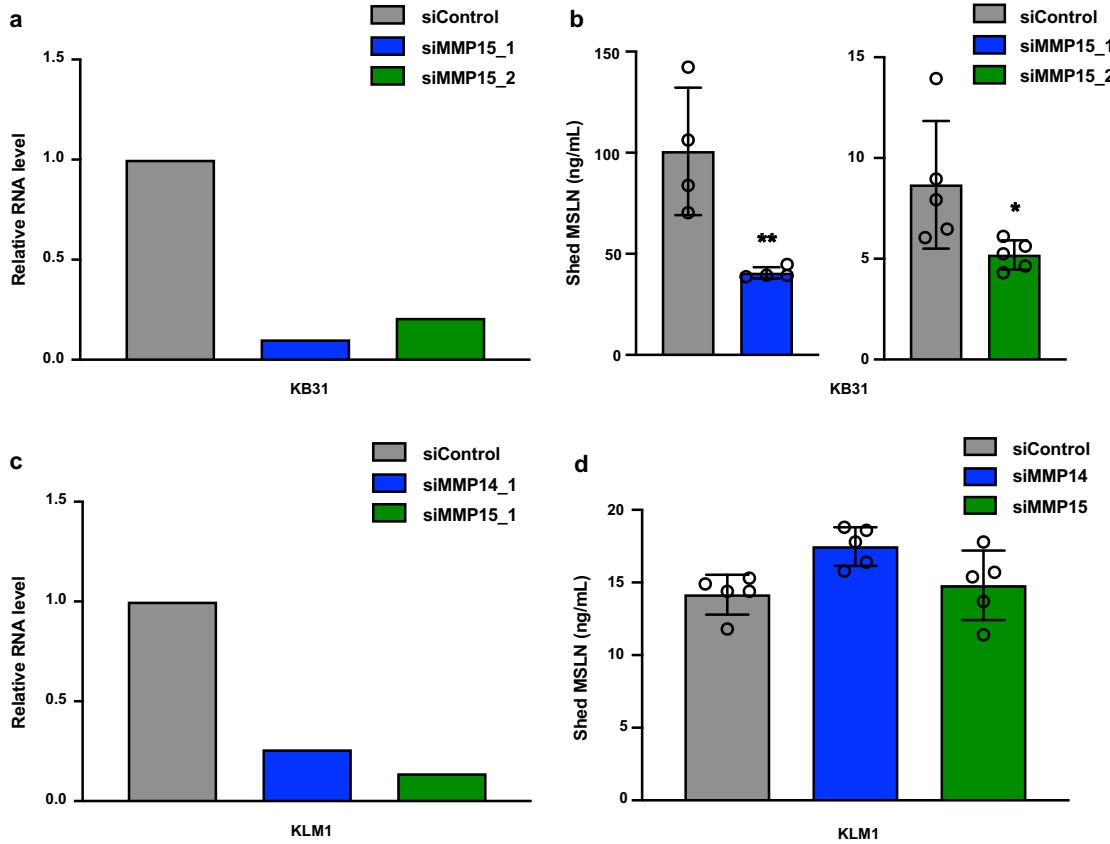

**Fig. 3 Knockdown of MMP15 decreased MSLN shedding in KB31 cells but not in KLM1 cells. a** KB31 cells were transfected with siRNA MMP15. After 48 h, the RNA was analyzed by real-time PCR. **b** Left, shed MSLN was measured 72 h after siRNA siMMP15_1 transfection in KB31 (**$P = 0.009$, $n = 4$). Right, shed MSLN was measured 48 h after siRNA siMMP15_2 (*$P = 0.043$, $n = 5$). **c, d** KLM1 cells were transfected with siRNA of siMMP14 and siMMP15 ($n = 5$). The relative RNA level (**c**) and shed MSLN (**d**) were similarly analyzed as in **a** and **b**.

and 2 in OVCAR8 but found no decrease in MSLN shedding (Supplementary Fig. 9). In summary, these data indicate that BACE2 but probably not BACE1 has a major role MSLN shedding.

**Small-molecule inhibitors to BACEs, ADAMs, and MMPs reduce MSLN shedding.** Lanabecestat (LY3314814) and LY2886721 are potent inhibitors of BACE1 and BACE2 that have been used in clinical trials[16,21,22]. We tested the inhibitors on cell lines where MSLN shedding was affected by BACE2 knockdown. Both drugs inhibited MSLN shedding by over 50% in KB31 and by 30–40% in MS751, T3M4, and AsPC1 cells (Fig. 6a). Neither agent inhibited MSLN shedding in KLM1 and OVCAR8 cells (Fig. 6a), which have very low BACE2 (Supplementary Table 2). We then measured the activities of the ADAM inhibitor TMI-1 and the MMP inhibitor Marimastat on several cell lines. Figure 6b shows that the TMI-1 greatly decreased MSLN shedding in KLM1, as expected from the knockdown experiments, and it also affected OVCAR8 and T3M4. Marimastat inhibited shedding in KB31, KLM1, T3M4, and OVCAR8 cells. Marimastat is a potent broad-spectrum inhibitor of all major MMPs[23]. We only knocked down membrane-bound MMPs; it is possible other MMPs play a role in MSLN shedding.

Because our knockdown studies showed both MMP15 and BACE inhibited shedding in KB31 cells, we tested inhibitors of these enzymes. Figure 6c shows that Lanabecestat inhibits shedding by 30%, that Marimastat lowers shedding by 50%, and the combination by 80%. This result shows shedding can be mediated by two enzymes in the same cell line. Furthermore, we suspect an additional sheddase is involved, since shedding was not completely inhibited. To determine if this inhibition would

result in enhanced immunotoxin action, we treated cells with the inhibitors for 20 h and then added SS1P, an immunotoxin that kills MSLN-expressing cells[1]. We found that the IC$_{50}$ changed from 4.6 ng/ml with no inhibitor to 1.1 ng/ml with Marimastat, to 1.3 ng/ml with Lanabecestat and to 0.29 ng/ml with both (Fig. 6d). The inhibitors were also examined in OVCAR8 cells. As shown in Fig. 6e, they stimulated SS1P killing; the IC$_{50}$ fell from 7.8 to 1.3 ng/ml with TMI-1, to 2.8 ng/ml with Marimastat and to 0.94 ng/ ml with both. These data show that in a single cell line more than one protease participates in MSLN shedding and inhibition greatly sensitizes the cells to an agent targeting MSLN.

**Do BACEs participate in shedding of other proteins.** The unexpected finding that BACEs have a major role in MSLN shedding raises the possibility that they would affect the release/ shedding of other proteins. To investigate this, we treated five cell lines with 20 µM Lanabecestat for 20 h and measured the content of several proteins and cytokines in the medium (Supplementary Table 3). As expected MSLN levels were decreased in all the lines, although the decrease in KLM1 was small and not significant.

There was also a small inhibition of shedding of Her1 (EGFR) in four lines and CA125 in one cell line (MS751). There was no effect on the folate receptor, which like MSLN is PI linked or on TGF-β1 and galectin 3. We conclude that the effect of BACEs on MSLN shedding is relatively specific.

**MBTPS1 and MEPRIN.** MBTPS1 has also been found to have a role in protein processing[16]. We knocked down MBTPS1 in KB31 and KLM1 cells and found no change in MSLN shedding

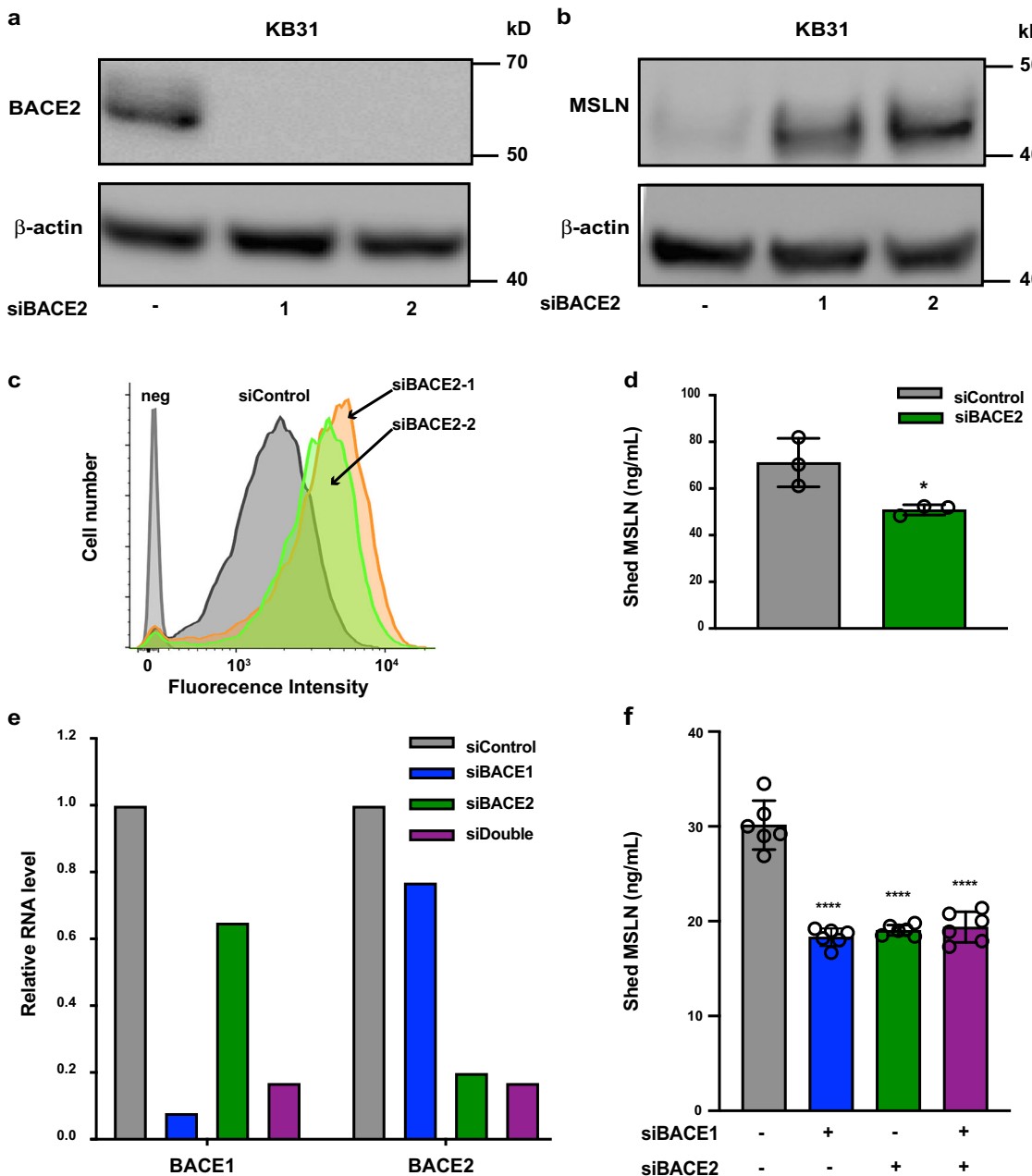

**Fig. 4 Knockdown of BACE lowers MLSN shedding in KB31 cells. a** KB31 cells were transfected with two different siRNA oligos targeting BACE2; after 72 h, cell lysates were analyzed by western blotting using anti-BACE2 antibody. **b** Cells were treated as in **a** and blotted with anti-MSLN antibody. Anti-actin is a loading control (full blots shown in Supplementary Fig. 7). **c** Cells were transfected with BACE2 siRNA, 72 h later labeled with anti-MSLN-Alexa647, and analyzed by flow cytometry. **d** Growth medium collected 48–72 h post-transfection of BACE2 siRNA was assayed for MSLN ($n = 3$, *$P = 0.03$). **e** Real-time PCR analysis of RNA levels 48 h after transfection of siBACE1 and siBACE2. **f** Shed MSLN level after transfection of BACE1 and BACE2 measured 48–72 h post-transfection ($n = 6$, ****$P < 0.00001$).

(Supplementary Fig. 10). Because there is no R residue in the region of MSLN where cutting occurs and cutting by MBTPS1 requires an R residue at P4, this result was expected. We also considered the possibility that meprin may participate in shedding, but CCLE shows that MEP1A and MEP1B RNA levels are very low in our cell line panel.

## Conclusion
We show here that MSLN shedding is an unexpectedly complex process, which involves many different enzymes that cleave

MSLN at several sites close to the cell membrane. We utilized several inhibitors that greatly decrease MSLN shedding and greatly enhance killing of cells by immunotoxins that target MSLN. Cutting at site 4 was only observed in KB31 cells, in which shedding is inhibited by BACE knockdown and the BACE inhibitor Lanabecestat, which did not inhibit MSLN shedding in KLM1 (Fig. 6a), OVAR8 (Fig. 6a) and RH16 (Supplementary Fig. 11). Cutting at site 3 is likely due to ADAM17, since it is found in cell lines sensitive to ADAM17 knockdown and inhibition by the ADAM inhibitor TMI-1. KB31 cells do not have cut site 3 and are not sensitive to TMI-1. We would expect these

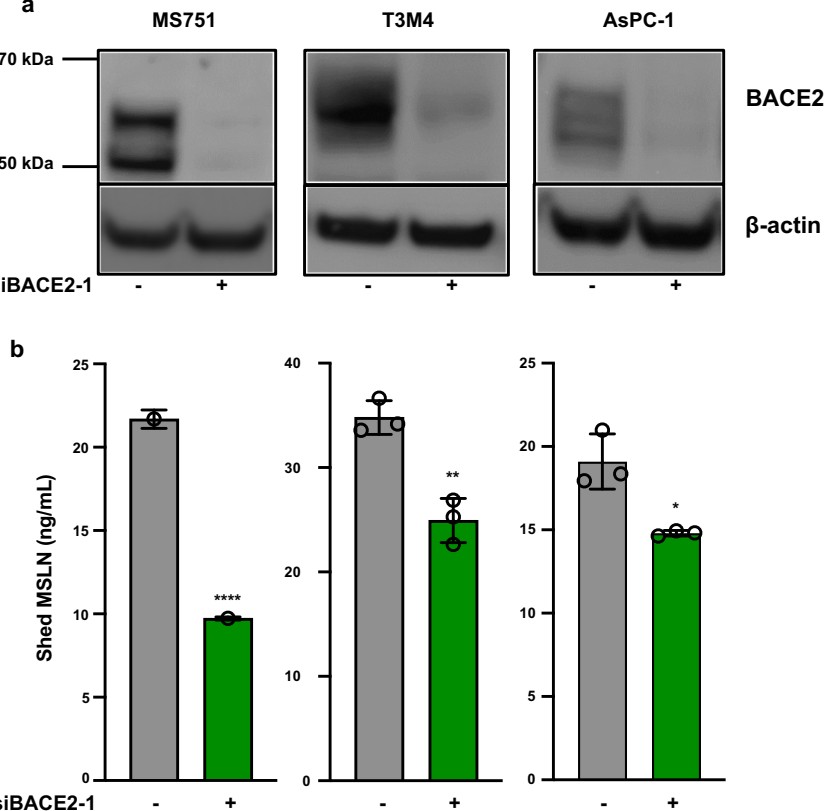

**Fig. 5 Knockdown of BACE2 in MS751, T3M4, and AsPC1 cell lines decreases MSLN shedding. a** Seventy-two hours after transfection of BACE2 siRNA into MS751, T3M4, and AsPC1, cell lysates were analyzed by western blotting using anti-BACE2 antibody (full blots shown in Supplementary Fig. 8). **b** Shed MSLN was collected 48–72 h after BACE2 siRNA. MS751($n = 3$, $P < 0.00001$), T3M4 ($n = 3$, $P = 0.0031$), AsPC1 ($n = 3$, $P = 0.01$). Statistical significance compared to a negative control siRNA is noted by asterisks (*$P < 0.05$, **$P < 0.01$, ****$P < 0.0001$).

inhibitors would also improve the activity of antibody drug conjugates, Chimeric antigen receptor (CARs), and other antibody-based therapies that are being developed to target cancers that express MSLN. Selective inhibitors of MSLN shedding could benefit patients receiving antibody-based therapies targeting MSLN-expressing tumors.

## Methods

**Materials**. Inhibitors LY2886721, Lanabecestat, and Marimastat were from Selleck Chemicals and TMI-1 was from Tocris Sciences. Anti-ADAM17 and anti-ADAM9 were from Cell Signaling Technology (Cat:3976S and 4151S, 1:000 dilution), anti-BACE2 antibody was from Santa Cruz Biotechnology (sc-271212, 1:1000 dilution), and anti-MSLN antibody (mouse monoclonal MN, 1 ug/ml) was from our lab. siRNAs against human ADAM17: s13719 (siADAM17_1) and si13720 (siA-DAM17_2) and one siRNA targeting ADAM10: s1004 were from Thermo Fisher Scientific. All other On-Target Plus siRNA and negative control siRNA luciferase GL2 were from Dharmacon.

**Cell culture**. KB31, KLM1, AsPC1, and A431/H9 cells were described before[24,25], T3M4 were from M. Ho (NCI). OVCAR8 and MS751 cells were purchased from ATCC. KB31, KLM1, T3M4, and AsPC1 were cultured in RPMI-1640 medium. MS751 cells were grown in EMEM medium. All culture media were supplemented with 10% FBS and 1% pen–strep. Cells were maintained at 5% $CO_2$ at 37 °C. All cells were tested regularly for mycoplasma contamination.

**Knockdown experiments**. All cells were transfected using Lipofectamine RNAi-MAX reagent according to the manufacturer's protocol in 6 well or 96 well. Either $5 \times 10e5$ cells were plated in each well of six-well plates, or 5000 cells were plated in each well of 96-well plates. After 24 h on plate, siRNAs were transfected at a final concentration of 60 nM in a volume of 2 ml or 100 µl, respectively. siRNA oligo sequences are listed in Supplementary Table 4.

**Soluble MSLN assay**. Cell culture media was collected after siRNA transfection or inhibitor treatment at indicated times. The culture media were centrifuged at 3000 r.p.m. table-top centrifuge for 10 min, diluted 5–10-fold in diluent #100 and assayed using the R-PLEX human MSLN antibody set following the manufacturer's instructions (Mesoscale Discovery, Rockville, MD). The electrochemiluminescence was measured on a MESO QuickPlex SQ instrument. Shed MSLN levels are highly dependent on cell density and growth conditions. In these studies, the ratio of cells to medium is higher in six-well plate format leading to higher MSLN levels, which varied from 5 to 50 ng/ml under different experimental conditions. The MSLN levels were usually normalized using WST-8 assays to correct for differences in cell growth.

**Real time PCR**. RNAs were made using the Triazole reagent (Invitrogen). Reverse transcription and cDNA synthesis used the Quantitect Reverse transcription kit following the manufacturer's instructions (Qiagen). Primers are listed in Supplementary Table 4. The PCR reaction was performed using Quantifast SYBR green PCR master kits (Qiagen).

**Western blotting**. Cell pellets were prepared in lysis buffer containing 50 mM Tris HCl, 150 mM NaCl, 5 mM EDTA with 1% NP40 and protease inhibitors. SDS-PAGE was run with an equal amount of protein on a NuPAGE 4–12% Bis-Tris gel under reducing conditions. Protein was transferred onto a PVDF membrane and subjected to western blot analysis using appropriate antibodies. Full blots are shown in Supplementary Figs. 1, 7, 8, 11 and 12.

**Detection of surface MSLN expression by flow cytometry**. KB31 cells were transfected with BACE2 siRNA for 48 h, harvested, and stained with MN antibody conjugated to AlexaFluor-647 and analyzed using a FACSCanto II flow cytometer. Data were analyzed using the FlowJo v10.4.2 software. Data were collected in duplicate and the experiment was repeated three times. Results are represented as mean ± SD from one experiment.

**Cytotoxicity assays**. In total, 10,000 cells were seeded in 96-well plates with various inhibitors at indicated concentration. After 20 h of treatment, serial

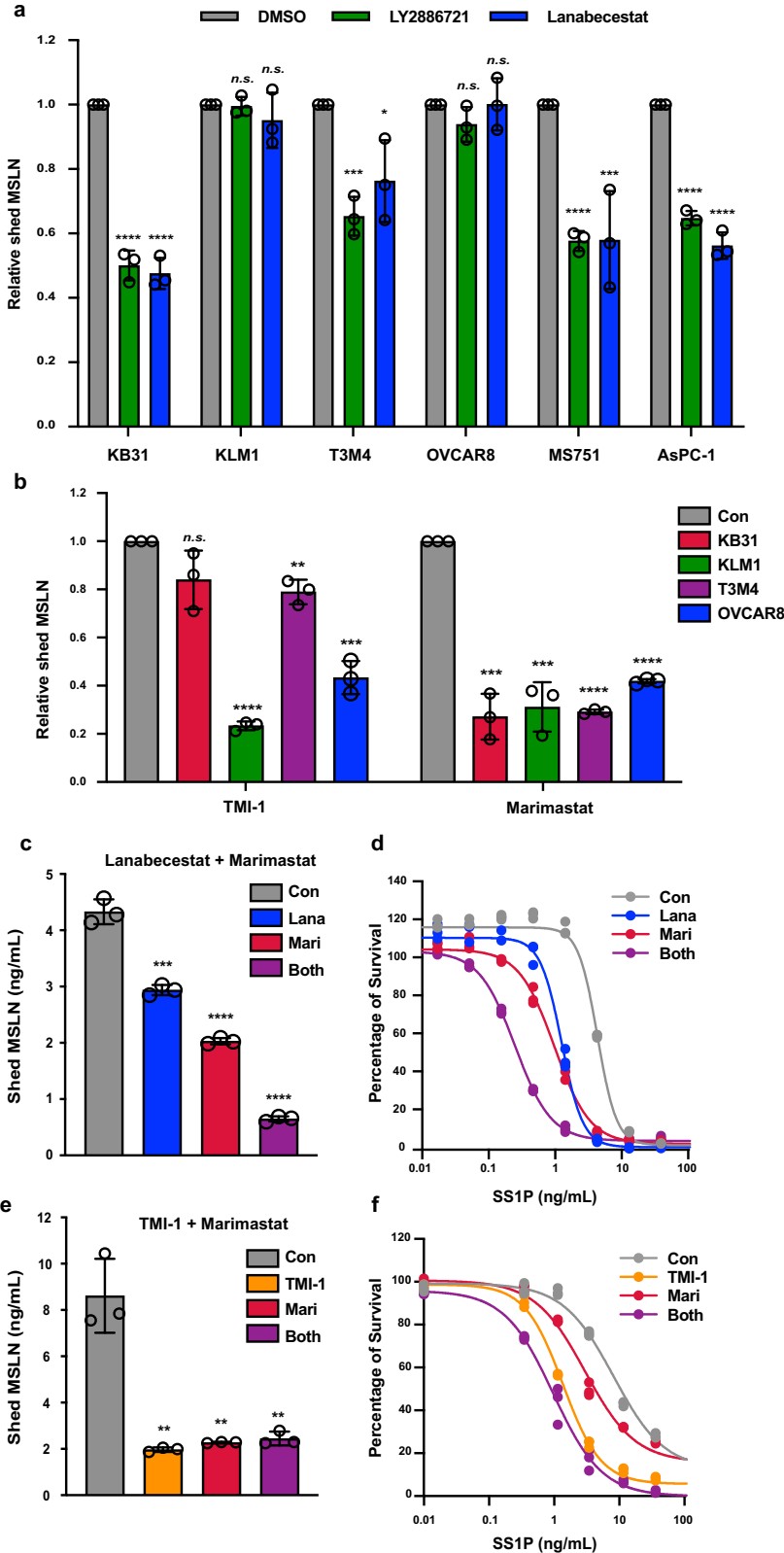

dilutions of SS1P were added in triplicate. Cells were incubated for an additional 72 h before running a WST-8 cell viability assay (Dojindo Molecular Technologies). Cell death was calculated with GraphPad Prism v7.05 relative to DMSO control samples. Viability is expressed as the percentage of reading at $OD_{450}$ with SS1P compared to control treatments without SS1P treatment.

**Gene expression analysis in cancer cell lines.** In searching for candidate MSLN sheddases, ADAMs, BACEs, and MMPs mRNA levels were compiled from cell lines with high MSLN expression. Expression data for OVCAR8, T3M4, AsPC1, and MS751 were downloaded from the Broad Institute Cancer Cell Line Encyclopedia (CCLE), and expression of genes in KB31 and KLM1 cells was determined by previous RNAseq experiments performed by our group.

**In gel trypsin digestion.** The Coomassie stained gel bands were chopped to small pieces and destained using 50% ACN/25 mM $NH_4HCO_3$, pH 8.4. After removal of organic solvent, the gel pieces were vacuum dried for 45 min. The dried gel pieces

**Fig. 6 Inhibitors of BACE, ADAM, and MMP lower MSLN shedding. a** Twenty micromolar of LY2886721 or Lanabecestat were incubated with KB31, MS751, T3M4, AsPC1, or KLM1 cells in 96-well plates for 48 h. MSLN in culture media was measured (KB31, both $P < 0.0001$; KLM1, $P = 0.77$ and 0.38; T3M4, $P = 0.0006$ and 0.031; OVCAR8, $P = 0.118$ and 0.987; MS751, $P < 0.0001$ and 0.009; AsPC1, both $P < 0.0001$, $n = 3$). **b** In all, 10 μM TMI-1 or 10 μM Marimastat were incubated with cells and shed MSLN measured (TMI-1, $P = 0.084$, <0.0001, 0.002, 0.00014; Marimastat: $P = 0.0002$, 0.0003, <0.0001, <0.0001, $n = 3$). **c–f** KB31 cells (**c, d**) were treated with 10 μM Lanabecestat or 10 μM Marimastat or both, OVCAR8 (**e, f**) were treated with TMI-1 or Marimastat or both. Shed MSLN level are shown (**c** $P = 0.0005$, <0.0001, <0.0001; **e** $P = 0.002$, 0.0023, 0.0027, $n = 3$). KB31 (**d**) or OVCAR8 (**f**) were incubated with inhibitors for 20 h, then SS1P at indicated concentration was added for 72 h. Cell viability was measured by WST-8 assay. Lana Lanabecestat, Mari Marimastat (ns not significant, *$P < 0.05$, **$P < 0.01$, ***$P < 0.001$, ****$P < 0.0001$).

were rehydrated in 50 μl of trypsin (20 ng/μl) resuspended in 25 mM NH$_4$HCO$_3$ and incubated on ice for 30 min. Excess trypsin was removed and replaced with 30 μl of 25 mM NH$_4$HCO$_3$, pH 8.4. The samples were incubated at 37 °C overnight. The peptides were extracted from the gel bands with 70% (v/v) ACN/0.1% TFA. The extracts were lyophilized, desalted using C18 columns (Thermo scientific), and reconstituted in 0.1% TFA.

**Mass spectrometry acquisition and data analysis**. The samples were injected on to Easy nLC 1000 HPLC (Thermo Scientific) coupled to an Orbitrap Fusion mass spectrometer (Thermo Scientific). The Easy nLC was configured with Acclaim Pepmap100 C18 2 cm trap column followed by 25 cm analytical column (Thermo Scientific) and the samples ran on a 90 min gradient at 250 nl/min. The precursor scans were performed in the Orbitrap at a resolution of 60,000 with a mass range of 375–1500 $m/z$, followed by MS/MS analysis at a resolution of 7500 in the ion trap. Normalized collison energy was 29, and charge state 1 and unassigned charge states were excluded. Acquired MS/MS spectra were searched against human Uniprot protein database or Mesothelin (Uniprot number Q13421 iso #1) using the SEQUEST algorithm in the Proteome Discoverer 2.2 (Thermo Scientific, CA). Enzyme was set to Trypsin (Semi), precursor ion tolerance was set at 10 ppm, and the fragment ions tolerance was set at 0.6 Da along with methionine oxidation included as dynamic modification. Peptide validation was done by either the Percolator or Target Decoy PSM Validator with FDR set at 0.1%.

**Statistics and reproducibility**. All data are presented as mean ± standard deviation (SD). The error bars represent the SD of triplicates or more as indicated in the figure legends. All experiments were repeated at 2–3 times. Statistical differences between groups were measured by unpaired $T$-test, two-tailed, using GraphPad Prism 7. A $P$ value of less than 0.05 was considered significant. All data are available upon request.

**Reporting summary**. Further information on research design is available in the Nature Research Reporting Summary linked to this article.

## Data availability

Original mass spectra has been deposited in MassIVE (https://massive.ucsd.edu) under the identifier MassIVE: MSV000085950. The source data behind graphs can be found in Source Data. All other data generated during and/or analyzed during the current study are available from the corresponding author on reasonable request.

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

## Acknowledgements

The authors thank Dr. Liang Cao for assistance with mesothelin ELISA assays and Drs. Oleg Chertov, Sudipto Das, and Andreasson Thorkell for sequence analysis. We thank Ms. Cynthia Hurlbert for editorial assistance. This research was supported by the Intramural Research Program of the NIH, NCI, CCR.

## Author contributions

I.P. designed research; X.L., A.C., and T.A. performed research; C.-H.T. provided analytic tools; X.L., A.C., T.A., and I.P. analyzed data; I.P., X.L., and A.C. wrote the paper.

## Competing interests

I.P. has patents to immunotoxins targeting mesothelin that have all been assigned to NIH. All other authors declare no competing interests.
