## [Peer Review File · Communications Biology]

Reviewers' comments:

Reviewer #1 (Remarks to the Author):

Liu et al describe a new mechanism of mesothelin (MSLN) shedding, the proteases involved, and the impact of MSLN shedding inhibition on the cytotoxicity of an immunotoxin. In particular, the authors describe 1) that members of the ADAM, MMP and BACE families of proteases participate in MSLN shedding, 2) that shedding occurs at different sites of MSLN close to the cell membrane, 3) that more than one protease can lead to MSLN shedding in the same cell, and 4) that knocking down/inhibiting specific proteases increases the efficacy of an immunotoxin targeting MSLN. The concept of MSLN shedding and its role in diminishing immunotoxin efficacy is not new and has been described by the same and other groups before. The novelty of the current manuscript is that it provides new insights into the complexity of MSLN shedding and how to overcome the diminishing effect via combination therapies. As MSLN constitutes a very promising target for solid tumors, different treatment modalities such as antibodies, antibody drug conjugates, and cell therapies such as CAR T cells are in development and currently being tested in various clinical trials. MSLN shedding can lead to diminished efficacy of antibody-based drugs as shown by the authors and, although not currently known, could potentially also affect cell therapy efficacy. The work by Liu et al is very valuable in that it not only improves our understanding of the mechanism of MSLN shedding but also provides guidance on how to overcome the diminishing effect of MSLN shedding on drug efficacy by exploring combination approaches of small molecule inhibitors together with a drug/immunotoxin targeting cancer cell surface-bound MSLN.

There are several, mostly minor points that should be addressed before publication:

- Levels of shed MSLN seem to vastly differ for the same cell line across different figures although culture time is 48-72hrs for all experiments, e.g. concentration of shed mesothelin for the cell line KLM1 ranges from ca. 20ng/mL in figure 2b to 60ng/mL in figure 2d and 3ng/mL in figure 2e. Were the culture conditions different between experiments?
- Lines 155-156: "We knocked down MBTPS1 in KB31 and KLM1 cells and found no change in MSLN shedding (Extended Data Fig. 7). Extended data figure 7 shows that the relative RNA level was not changed for siRNA treatment but shed MSLN level was ca. 4-fold decreased. Please explain.
- Why do the authors sometimes see an increase in MSLN shedding upon protease knockdown, e.g. extended data figure 2d, 3b, 4b?
- Figure 6b: How do the authors explain Marimastat has an effect on MSLN shedding in KLM1 and OVCAR1 cells that was not seen upon knockdown of MMP14/15?
- Line 147: "As expected MSLN levels were significantly decreased in all the lines with the smallest decrease in KLM1 [...]". According to extended data table 2 cell line KLM1 is not significantly different.
- What are the references in lines 356 to 361 referring to? Should be included in general reference section.
- Extended data table 1: It should be made more clear that the color coding (bold and red) is derived from the results of the manuscript as it could be misinterpreted that this data is also derived from CCLE . Unit RPKM should be explained.
- Extended data table 1 and 2: Cell line "KB" should be "KB31".
- Extended data table 1: It would be nice to include cell lines RH16 and A431 as well.
- Line 104: "Since BACE1 RNA is also expressed in KB31 cells, albeit at a lower level than BACE2, [...]Please include that this is based on Extended Data Table 1.
- Please harmonize the names for the cell lines (e.g. OVCAR8 or OVCAR-8, ASPC1 or AsPC1 or AsPC-1).
- Please include antibody clones used in the materials section.

Reviewer #3 (Remarks to the Author):

The authors present novel work in which they found that at least three protease families contribute to shedding of Mesothelin from the surface of cancer cells. They further demonstrate that inhibition of shedding increases cell death induced by Mesothelin targeted agents. The study is a valuable contribution to the ongoing discussion of targeted cancer therapies, resistance and combination therapies. The presented work includes potentially interesting and relevant findings, however given the details provided and low level of replication it remains unclear how robust the findings are.

Mayor Comments.

- My main concern lies with the questionable robustness and reproducibility of the data. As per the authors statement only two or three datapoints were acquired per experiment and based on some graphs that do not have error-bars I assume that some experiments may even be one offs. Given that the conducted experiments are not overly complex, time-consuming and expensive the current norm would be to have at the very least 5 replica, ideally more but certainly not 2.
- The description of the statistical approaches is incomplete (e.g. what kind of t test).
- Bar graphs are not a suitable representation for the data. At the very least all datapoints need to be shown as well.
- The authors start off discussing the different cleavage sites found in MSLN and hypothesize that these indicate different protease activities. With this lead off and interesting hypothesis I would have expected that the authors in the following not only determine bulk change in shedding but also evaluate which particular cleavage site is affected by a given knock down or inhibitor treatment.
- MS data supporting the identified C-terminal sequences of MSLN is missing. This needs to be provided and also deposited with PRIDE.
- The methods section is lacking a lot of relevant details (e.g. concentrations, media changes) and the section describing the GC-MS analysis is missing entirely. It would be impossible to reproduce the data with the information given.

Minor comments and suggested improvements:

- On page 3 the authors state that "A431 cells do not express MSLN" while on page 4 they state that "ADAM17 was previously shown to be important for shedding of MSLN in A431/H9 cells ". This contradiction needs to be resolved and ideally supported by data.
- Page 5: "However, knocking down both did not further lower MSLN shedding, raising the possibility that when one enzyme is knocked down, the other for its loss." . Wouldn't one conclude the exact opposite?

Reviewer #1 (Remarks to the Author):

Liu et al describe a new mechanism of mesothelin (MSLN) shedding, the proteases involved, and the impact of MSLN shedding inhibition on the cytotoxicity of an immunotoxin. In particular, the authors describe 1) that members of the ADAM, MMP and BACE families of proteases participate in MSLN shedding, 2) that shedding occurs at different sites of MSLN close to the cell membrane, 3) that more than one protease can lead to MSLN shedding in the same cell, and 4) that knocking down/inhibiting specific proteases increases the efficacy of an immunotoxin targeting MSLN. The concept of MSLN shedding and its role in diminishing immunotoxin efficacy is not new and has been described by the same and other groups before. The novelty of the current manuscript is that it provides new insights into the complexity of MSLN shedding and how to overcome the diminishing effect via combination therapies. As MSLN constitutes a very promising target for solid tumors, different treatment modalities such as antibodies, antibody drug conjugates, and cell therapies such as CAR T cells are in development and currently being tested in various clinical trials. MSLN shedding can lead to diminished efficacy of antibody-based drugs as shown by the authors and, although not currently known, could potentially also affect cell therapy efficacy. The work by Liu et al is very valuable in that it not only improves our understanding of the mechanism of MSLN shedding but also provides guidance on how to overcome the diminishing effect of MSLN shedding on drug efficacy by exploring combination approaches of small molecule inhibitors together with a drug/immunotoxin targeting cancer cell surface-bound MSLN.

There are several, mostly minor points that should be addressed before publication:

- Levels of shed MSLN seem to vastly differ for the same cell line across different figures although culture time is 48-72hrs for all experiments, e.g. concentration of shed mesothelin for the cell line KLM1 ranges from ca. 20ng/mL in figure 2b to 60ng/mL in figure 2d and 3ng/mL in figure 2e. Were the culture conditions different between experiments?

Yes, shed MSLN levels are dependent on cell density and growth conditions. In Fig 2b and Fig 2d, the cells were transfected in 6 well plates and the starting cell number is 5×10^5 cells. In Fig 2e, the cells were in 96 well plates and the starting cell number is 5000 cells in 100 μ l media.

We added this explanation in the Material and Methods section line 321 and lines 325-328.

“Shed MSLN levels are highly dependent on cell density and growth conditions. In these studies, the ratio of cells to medium is higher in 6-well plate format leading to higher MSLN levels, which varied from 5-50 ng/ml under different experimental conditions.”

- Lines 155-156: “We knocked down MBTPS1 in KB31 and KLM1 cells and found no change in MSLN shedding (Extended Data Fig. 7). Extended data figure 7 shows that the relative RNA level was not changed for siRNA treatment but shed MSLN level was ca. 4-fold decreased. Please explain

The figure labels were mistakenly reversed, MSLN shedding and RNA level were oppositely designated. We have corrected the axis labels.

- Why do the authors sometimes see an increase in MSLN shedding upon protease knockdown, e.g. extended data figure 2d, 3b, 4b?

We do see shed MSLN increase after knocking down sheddases in some experiments. We assume this is due to upregulation of other sheddases, cells may compensate for the loss of knockdown.

- Figure 6b: How do the authors explain Marimastat has an effect on MSLN shedding in KLM1 and OVCAR1 cells that was not seen upon knockdown of MMP14/15?

“Marimastat is a potent broad-spectrum inhibitor of all major MMPs. We only knocked down membrane bound MMPs, other MMPs may also play a role in MSLN shedding.” We added this sentence in the text line 125-127.

- Line 147: “As expected MSLN levels were significantly decreased in all the lines with the smallest decrease in KLM1 [...]”. According to extended data table 2 cell line KLM1 is not significantly different.

We apologize for the mistake, the sentence should state “As expected MSLN levels were decreased in all the lines, with the smallest decrease in KLM1, which is not significant. We put this in the text in lines 146-147.

- What are the references in lines 356 to 361 referring to? Should be included in general reference section.

The references were for the Materials and Methods. The numbering was incorrect, but has been fixed per the guidelines from Nature, which asks for them to be listed after the section, but with sequential numbering.

- Extended data table 1: It should be made more clear that the color coding (bold and red) is derived from the results of the manuscript as it could be misinterpreted that this data is also derived from CCLE. Unit RPKM should be explained.

These are good points. We added the table legend, starting with line 451.

- Extended data table 1 and 2: Cell line “KB” should be “KB31”.

This was corrected.

- Extended data table 1: It would be nice to include cell lines RH16 and A431 as well.

This data has been added.

- Line 104: “Since BACE1 RNA is also expressed in KB31 cells, albeit at a lower level than BACE2, [...]Please include that this is based on Extended Data Table 1.

Thanks for the suggestion, we put it in line 104.

- Please harmonize the names for the cell lines (e.g. OVCAR8 or OVCAR-8, ASPC1 or AsPC1 or AsPC-1).

Yes, we made all the names as OVCAR8 and AsPC1.

- Please include antibody clones used in the materials section.

We added the antibody Catalog or clone name in line 299-301.

Reviewer #3 (Remarks to the Author):

The authors present novel work in which they found that at least three protease families contribute to shedding of Mesothelin from the surface of cancer cells. They further demonstrate that inhibition of shedding increases cell death induced by Mesothelin targeted agents. The study is a valuable contribution to the ongoing discussion of targeted cancer therapies, resistance and combination therapies. The presented work includes potentially interesting and relevant findings, however given the details provided and low level of replication it remains unclear how robust the findings are.

Major Comments.

- My main concern lies with the questionable robustness and reproducibility of the data. As per the authors statement only two or three datapoints were acquired per experiment and based on some graphs that do not have error-bars I assume that some experiments may even be one offs. Given that the conducted experiments are not overly complex, time-consuming and expensive the current norm would be to have at the very least 5 replica, ideally more but certainly not 2.

All the data presented in the figures are not single points. Western blots, RNA levels and cytotoxicity assays were repeated at least twice and mesothelin assays were repeated 3 or more times and a representative experiment is chosen for each experiment. Most data shown is the mean of triplicates, but some are duplicates, Fig.3b and 3d, Fig. 4d, Fig. 5. (T3M4).

- The description of the statistical approaches is incomplete (e.g. what kind of t test).

We now modified the statistical approach as follows and appeared from line 367. All data are presented as mean \pm standard deviation (SD). The error bars represent the SD of triplicate except Fig.3b and 3d, Fig. 4d, Fig. 5 (T3M4) which are duplicates. The experiments were repeated at least 3 times. Statistical differences between groups were measured by Multiple

t test-One per row using GraphPad Prism 7. A *P* value of less than 0.05 was considered significant.

- Bar graphs are not a suitable representation for the data. At the very least all datapoints need to be shown as well.⁷

Since all the points are very close as seen by the error bar, we do not plan to change the graph.

- The authors start off discussing the different cleavage sites found in MSLN and hypothesize that these indicate different protease activities. With this lead off and interesting hypothesis I would have expected that the authors in the following not only determine bulk change in shedding but also evaluate which particular cleavage site is affected by a given knock down or inhibitor treatment.

We expanded the discussion to cover this important point, line 164-169.

Cutting at site 4 was only observed in KB31 cells, in which shedding is inhibited by BACE knock down and the BACE inhibitor, Lanabecestat, which did not inhibit MSLN shedding in KLM1 (Fig. 6a), OVCAR8 (Fig. 6a) and RH16 (Extended data Figure 8). Cutting at site 3 is likely due to ADAM17, since it is found in cell lines sensitive to ADAM17 knock down and inhibition by the ADAM inhibitor TMI-1. KB31 cells do not have cut site 3 and are not sensitive to TMI-1.

- MS data supporting the identified C-terminal sequences of MSLN is missing. This needs to be provided and also deposited with PRIDE.

MS data supporting the C-terminal sequence of MSLN will be added in extended table 4. It was also added in line 364. The data will be deposited once the paper accepted.

- The methods section is lacking a lot of relevant details (e.g. concentrations, media changes) and the section describing the GC-MS analysis is missing entirely. It would be impossible to reproduce the data with the information given.

We modified the material and methods and added concentrations, media change in line 315 and line 321. We also added the sentence to line 364, GC-MS analysis was performed as previously described (6).

Minor comments and suggested improvements:

- On page 3 the authors state that “A431 cells do not express MSLN” while on page 4 they state that “ADAM17 was previously shown to be important for shedding of MSLN in A431/H9 cells “. This contradiction needs to be resolved and ideally supported by data.

A431 is a cell line that does not express MSLN, line 246. A431/H9 is a MSLN transfected stable cell line; this was added in line 247. Please also see Ref. #6

- Page 5: “However, knocking down both did not further lower MSLN shedding, raising the possibility that when one enzyme is knocked down, the other for its loss.”. Wouldn’t one conclude the exact opposite?

We agree, it is confusing, so we deleted the sentence in line 107.

Reviewers' comments:

Reviewer #1 (Remarks to the Author):

The authors addressed all points in a well-thought manner and made appropriate revisions to the manuscript. The only comment I have is that Table 4 is currently redundant to Figure 1 but as mentioned by the authors, Table 4 would be complemented with MS data when the paper is accepted.

Reviewer #2 (Remarks to the Author):

The authors have addressed all points raised by reviewers 1 and 2 in form of concise additions or corrections in the text. A response to reviewer 3 was missing in my version.

While the clarifications and additions do address some of the points, my main concern of reproducibility remains. The number of replica is low throughout (2-3) and as such render the presented results nice preliminary findings that require further confirmation before publication. Along the same lines the revised version still does not contain any detailed mass spectrometry results. The newly added extended data table 4 contains only sequences without any primary mass spec data. The authors state that raw data will only be deposited upon acceptance of the manuscript. The experimental details for the mass spectrometry experiments now refer to a paper (referenced only in the response to reviewers but reference missing in the manuscript) that presents MALDI data, but no GC-MS data supposedly used in this study. As such the quality and validity of this part of the manuscript can still not be assessed.

Details on the statistical test applied remain incomplete. Was the test one or two tailed, paired or unpaired? Also, even though designed for small sample sizes, it is questionable to apply a student's t test to two samples per group.

Minor point: The actin bands in extended figure 3 show a downward curvature unlike any other actin bands in the manuscript. This figure segment might have been inadvertently rotated or mirrored. Information on the number of replica has now been added but remains buried in the methods section. Ideally it would be stated in the figure legend to be easily accessible. For some figures only 'representative' samples are shown. Here the full data should be plotted.

Dear Editors,

Please find below our responses to the reviewers.

Reviewers' comments:

Reviewer #1 (Remarks to the Author):

The authors addressed all points in a well-thought manner and made appropriate revisions to the manuscript. The only comment I have is that Table 4 is currently redundant to Figure 1 but as mentioned by the authors, Table 4 would be complemented with MS data when the paper is accepted.

This is correct.

Reviewer #2 (Remarks to the Author):

The authors have addressed all points raised by reviewers 1 and 2 in form of concise additions or corrections in the text. A response to reviewer 3 was missing in my version.

While the clarifications and additions do address some of the points, my main concern of reproducibility remains. The number of replica is low throughout (2-3) and as such render the presented results nice preliminary findings that require further confirmation before publication.

We repeated all the 2-samples figures including Fig 3b, 3d, Fig 4d and Fig 5 (T3M4), and the figures were replaced with the new data. All the figures represent sample size equal or more than 3 as indicated in the figure legends.

Fig 3

Line 269, Fig 3 b, Left, shed MSLN was measured 72 hr after siRNA siMMP15_1 transfection in KB31 (** $P=0.009$, $n=4$). Right, shed MSLN was measured 48hr after siRNA siMMP15_2 (* $P=0.043$, $n=5$).

Line 271, Fig 3d: KLM1 cells were transfected with siRNA of siMMP14 and siMMP15_1 ($n=5$, $P=0.0048$ for siMMP14 or $P=0.618$ for siMMP15_1)

Fig 4

Line 280, Fig 4d: **d**, Growth medium collected 48-72 hr post-transfection of BACE2 siRNA was assayed for MSLN ($n=3$, * $P=0.03$).

Fig 5

Line 288, Fig 5 b, Shed MSLN was collected 48-72 hr after BACE2 siRNA. MS751 ($n=3$, $p=0.0001$); T3M4 ($n=3$, $p=0.0031$); AsPC-1 ($n=3$, $p=0.01$).

Along the same lines the revised version still does not contain any detailed mass spectrometry results. The newly added extended data table 4 contains only sequences without any primary mass spec data. The authors state that raw data will only be deposited upon acceptance of the manuscript. The experimental details for the mass spectrometry experiments now refer to a paper (referenced only in the response to reviewers but reference missing in the manuscript) that presents MALDI data, but no GC-MS data supposedly used in this study. As such the quality and validity of this part of the manuscript can still not be assessed.

We added detailed analysis of LC-MS in the method section: In gel trypsin digestion and Mass Spectrometry acquisition and data analysis from page 373 -396. The Original mass spectra has been deposited in MassIVE (<https://massive.ucsd.edu>) under the identifier MassIVE: MSV000085950.

Details on the statistical test applied remain incomplete. Was the test one or two tailed, paired or unpaired? Also, even though designed for small sample sizes, it is questionable to apply a student's t test to two samples per group.

We added the detail of statistics in line 401, Statistical differences between groups were measured by unpaired T-test, two-tailed, using GraphPad Prism 7.

Minor point: The actin bands in extended figure 3 show a downward curvature unlike any other

actin bands in the manuscript. This figure segment might have been inadvertently rotated or mirrored. Information on the number of replica has now been added but remains buried in the methods section. Ideally it would be stated in the figure legend to be easily accessible. For some figures only 'representative' samples are shown. Here the full data should be plotted.

Extended Data Fig 3

We corrected tilted actin band and make it straight in the extended figure 3a. All the information on the number of replicates has been added in the figures and figure legends.

REVIEWERS' COMMENTS:

Reviewer #2 (Remarks to the Author):

I would like to thank the authors for taking the time and invest the additional resources required to increase the number of replica throughout their main study. This adds considerable credibility to their interesting findings.

The majority of my concerns have been addressed. Some minor aspects to note, that would be easy to fix:

- For many experiments the time after which the amount of shed MSLN1 has been measured is stated as a range of 48-72 hours. The newly added Fig 3b shows that even in control cells the amount shed at 48 and 72 hours varies about 10 fold. In light of this it would be important to state a precise time rather than a span for each experiment and ensure that controls and treated samples have been collected after the same time.
- While the number of replica has been increase to 2-5 in the main figures some extended data figures still appear to have lower number of replica or be one offs. This is somewhat acceptable for extended figures, however the N should be clearly stated in the figure legend.
- The p-value for siMMP14 in Fig 3d is surprisingly low and should be checked for correctness.
- The actin bands in extended data figure 3 remain to have inverse shape compared to the actin bands in all other figures. My concern was not the tilt of the bands across the gel (which the authors corrected) but the shape of the individual band that is n shaped rather than u shaped - however this might just be the way these particular gels turned out.

I did not re-evaluate the rest of the manuscript in light of the newly added data.

Reviewer #2 (Remarks to the Author):

I would like to thank the authors for taking the time and invest the additional resources required to increase the number of replica throughout their main study. This adds considerable credibility to their interesting findings.

The majority of my concerns have been addressed. Some minor aspects to note, that would be easy to fix:
- For many experiments the time after which the amount of shed MSLN1 has been measured is stated as a range of 48-72 hours. The newly added Fig 3b shows that even in control cells the amount shed at 48 and 72 hours varies about 10 fold. In light of this it would be important to state a precise time rather than a span for each experiment and ensure that controls and treated samples have been collected after the same time.

We always collected control and treated samples at the same time. The 48hr-72hr span is the time when fresh media was incubated with the cells after transfection. The relative MSLN level is related to cell density as stated in the Material and Methods line 215-218.

- While the number of replica has been increase to 2-5 in the main figures some extended data figures still appear to have lower number of replica or be one offs. This is somewhat acceptable for extended figures, however the N should be clearly stated in the figure legend.

Thanks for the suggestion, we added the N number in the supplemental figure legends.

- The p-value for siMMP14 in Fig 3d is surprisingly low and should be checked for correctness.

We double checked the P value using GraphPad Prism 7 for Fig 3d correctness. However, we think it is better to remove it from the figure legends, because it did not show the siMMP14 is lower than the control, line 402-403.

- The actin bands in extended data figure 3 remain to have inverse shape compared to the actin bands in all other figures. My concern was not the tilt of the bands across the gel (which the authors corrected) but the shape of the individual band that is n shaped rather than u shaped - however this might just be the way these particular gels turned out.

We have corrected the gel image; the original images were presented in the Supplemental figure 9-13.

Note:

As we reformatted all the figures to fit the journal requirements, we replotted all the graphs using the original dots, and recalculated the P values. We made a correction on Figure 6b; we previously found that the inhibitor TMI-1 did not significantly affect T3M4, but on re-calculation we find the level is lower and the P value is significant. This change was described in the text, line 134.

2nd change: we previously used ***P<0.001 as highest significance. Now we added ****P<0.0001 to make value consistent with the convention of GraphPad Prism7. All the P value and N numbers are written in the figure legends.